# Non-Physical Disease Facets in Spondyloarthritis: An ASAS Health Index-Based Analysis between Psoriatic Arthritis and Axial Spondyloarthritis

**DOI:** 10.3390/jcm11206094

**Published:** 2022-10-16

**Authors:** Rubén Queiro, Sara Alonso, Isla Morante, Mercedes Alperi

**Affiliations:** 1Rheumatology Division, Hospital Universitario Central de Asturias, 33011 Oviedo, Spain; 2ISPA Translational Immunology Division, 33011 Oviedo, Spain; 3School of Medicine, Oviedo University, 33003 Oviedo, Spain; 4Rheumatology Division, Hospital de Sierrallana, 39300 Torrelavega, Spain

**Keywords:** axial spondyloarthritis, psoriatic arthritis, quality of life, ASAS health index

## Abstract

Background: Psychosocial health is a key driver of quality of life (QoL) in axial spondyloarthritis (axSpA) and psoriatic arthritis (PsA), but it is often overlooked in clinical practice. We aimed to analyze this aspect of QoL by using the Assessment of SpA International Society–Health Index (ASAS HI) in both SpA phenotypes. Patients and methods: One hundred and eleven patients with axSpA and 90 with PsA were consecutively recruited from two rheumatology centers. In both populations, the categories of stress handling (ASAS HI items #11 and 17) and emotional functions (ASAS HI item #13) were analyzed based on the International Classification of Functioning, Disability, and Health (ICF). A multivariate regression model was used to analyze the explanatory factors associated with positive responses to these items. Results: Thirty-four of the 90 PsA patients (37.8%) and 37/111 of the patients (33.3%) with axSpA reported a positive response to at least one of the stress-handling items. Compared to the patients with PsA, patients with axSpA were less likely to report stress-handling issues (OR 0.48, *p* < 0.05). Thirty-one of the 90 PsA patients (34.4%) and 44/111 of the patients (39.6%) with axSpA reported positive responses to item #13. In both groups of SpA patients, disease activity and severity (OR 6.6, *p* < 0.001) were independently associated with alterations in psychosocial health. Compared with those in the axSpA group, the psychosocial health items were better correlated with each other and with the ASAS HI sum score in the PsA group. Conclusions: Psychosocial health is frequently altered in SpA. Both disease activity and severity are associated with this issue. However, psychosocial factors seem to have a greater impact on QoL in PsA than in axSpA.

## 1. Introduction

Spondyloarthropathies (SpAs) are entities that limit the physical capacities of patients, in addition to altering their quality of life (QoL). However, these entities not only affect the physical spheres of patients, but also the psychosocial ones [1]. In a recent global survey of more than 1200 patients with psoriatic arthritis (PsA) from eight countries, in addition to physical impairment, 69% of patients reported that their disease had a moderate/major impact on their emotional/mental wellbeing, 56% reported such an impact on their romantic relationships/intimacy, and 44% reported such an impact on their relationships with family and friends. This was despite 56% of the patients currently receiving biological therapy [2]. An international survey on axial spondyloarthritis (axSpA) reported that 61.5% of patients were at risk for psychological distress, with 33.8% and 38.6%, respectively, reporting depression and anxiety [3]. A recent systematic review and meta-analysis of epidemiological studies of depression and anxiety in patients with PsA revealed a pooled OR of 1.68 for depression and 1.49 for anxiety compared to non-PsA populations [4]. These data highlight the significant burden of these psychosocial factors on the health-related QoL (HRQoL) in both types of SpA. Furthermore, these patients may be frustrated by delayed diagnoses and a lack of communication with their healthcare providers [5,6].

These non-physical disease facets are less commonly addressed in clinical routines, so it is necessary to have feasible instruments that capture the true impacts that these conditions generate in the day-to-day lives of these patients. Currently, both the PsA Impact of Disease (PsAID) questionnaire and the Assessment of SpA International Society–Health Index (ASAS HI) seem to be useful tools in this regard. However, the former is specific for PsA, while the latter can be applied to any type of SpA [7,8].

In patients with axSpA, physical domains are usually more affected than emotional ones, whereas for PsA, both physical and psychological factors are strongly affected by the disease [1,9]. The decision making for SpA should be a shared process between doctors and patients. Therefore, as patients with axSpA and PsA may present differences in the ways in which they perceive the impact of their disease, it would be of interest to have disease-impact-related comparisons to establish management strategies that are adapted to each type of SpA [10]. The objective of this ASAS-HI-based study was to analyze differences in psychosocial health between patients with PsA and those axSpA.

## 2. Materials and Methods

This is a post hoc comparative analysis of two studies in which we verified the construct and discriminant validity of the ASAS HI for axSpA and PsA. The methodological details, main results, and ethical considerations, which are applicable to both studies, have been published elsewhere [11,12]. Briefly, detailed data on the family history of the disease, educational level, comorbidities, received treatments, analytical data, structural damage, disease sub-phenotypes, and several composite outcome measures were collected. The latter were used as external anchors to validate the psychometric properties of the ASAS HI in both subpopulations [11,12].

As an outcome measure of disease impact, we used the ASAS HI. The ASAS HI is a linear composite measure based on the International Classification of Functioning, Disability, and Health (also known as ICF) containing 17 items that address aspects of pain, emotional functions, sleep, sexual functions, mobility, self-care, and community life. Each positive answer is scored as 1, while a negative answer is scored as 0. The final score is the sum of the individual items, so the higher the score, the greater the negative impact due to the disease [8]. The dependent variables in this secondary analysis were items #11 (“I find it hard to concentrate”) and #17 (“I cannot overcome my difficulties”), with both referring to the stress-handling ICF category, as well as item #13 (“I often get frustrated”), which refers to the ICF category of emotional functions [8].

### Statistical Analysis

A descriptive statistical analysis of all variables was performed, including the central tendency and dispersion measures for the continuous variables and absolute and relative frequencies for the categorical variables. The distribution of the components of the ASAS HI was compared between both diseases. Pearson’s r was used to analyze the correlations between items #11, #17, and #13 with the sum score of the ASAS HI, as well as those between these items and each individual component of this questionnaire. The correlation matrices of the items of the ASAS HI for both diseases have been published elsewhere [11,12]. A multivariate regression model (including age, gender, disease duration, and systemic therapy as covariates) was developed to analyze the explanatory factors associated with a positive statement about these items. Statistical significance was set at a value below 5%. The data were analyzed using R Statistical Software.

## 3. Results

### 3.1. Summary of the Study Population

This analysis included 111 consecutive patients with axSpA (median age: 42 years (IQR: 36–50), 74 men, 37 women, median duration of disease: 5 years (IQR: 4–10)) and 90 consecutive patients with PsA (median age: 53 (IQR: 45–65), 52 men, 38 women, median duration of arthritis: 7 years (IQR: 3–14), median psoriasis duration of 16 years (IQR: 10–29)). Table 1 shows the main disease features of both populations.

The mean ASAS-HI scores were 5.8 ± 4.4 in the PsA group and 5.4 ± 3.8 in the axSpA group. The ASAS HI and PsAID values were highly correlated (r: 0.75, 95% CI: 0.64–0.83). Compared to patients with axSpA, patients with PsA were older (55.3 ± 14.3 years vs. 43.3 ± 10.6 years, *p* < 0.001) and received fewer biological therapies in the inclusion visit (44.4% vs. 55.9%, *p* < 0.05); however, more PsA subjects were in remission/low activity upon study entry (73% vs. 48.6%, *p* < 0.001).

### 3.2. Stress Handling

In subjects with PsA, 30/90 (33%) responded affirmatively to item #11, while 31/111 (27.9%) of the axSpA patients reported concentration difficulties in response to this ASAS HI item. A moderate correlation (r: 0.62 in PsA and 0.60 in axSpA) was found between item #11 and the sum score of the ASAS HI. In the multivariate analysis, current use of biological therapy (OR 0.27, 95% CI: 0.06–0.95, *p* < 0.05) and disease activity according to DAPSA (OR 1.30, 95% CI: 1.17–1.50, *p* < 0.001) were independently associated with a positive response to item #11 in patients with PsA. In the axSpA patients, the factors independently related to a positive response to item #11 were biological therapy (OR 4.2, 95% CI:1.06–20.6, *p* < 0.05), depression (OR 40.9, 95% CI: 3.1–1215.8, *p* = 0.01), and disease activity according to BASDAI (OR 3.9, 95% CI: 1.4–15.8, *p* = 0.02). Taking both populations together, a severe disease (BASDAI > 4 + ASDAS-CRP > 2.1 + DAPSA > 14 + physician’s global disease assessment > 4) was independently related to a positive response to item #11 (OR 4.6, 95% CI: 2.3–9.3, *p* < 0.001).

With respect to item #17, 17/90 (18.9%) of the PsA patients reported a positive response to this item compared to 15/111 (13.5%) of the axSpA patients. There were greater correlations between item #17 and the sum score of the ASAS HI in the PsA group (r: 0.65) than in the axSpA group (r: 0.53). In the PsA group, the only factor independently associated with a positive response to item #17 was disease activity according to DAPSA (OR 1.28, 95% CI: 1.15–1.48, *p* < 0.001). In addition, the disease activity according to ASDAS was related to this item in the axSpA group (OR 79.3, 95% CI: 8.9–2550.2, *p* = 0.002). Taking both populations together, a severe disease (BASDAI > 4 + ASDAS-CRP > 2.1 + DAPSA > 14 + physician’s global disease assessment > 4) was independently related to a positive response to this item (OR 15.8, 95% CI: 5.7–52.4, *p* < 0.001).

Of the patients with PsA, 34/90 (37.8%) answered positively to at least one of the two items referring to stress handling. Among the patients with axSpA, 37/111 (33.3%) reported a positive response to at least one of the two items. Taking this into account, the arthritis duration (OR 1.09, 95% CI: 1.02–1.18, *p* = 0.02) and disease activity measured with DAPSA (OR 1.32, 95% CI: 1.18–1.51, *p* < 0.001) were independently related to a positive response to the stress-handling items. In the same scenario, the factors associated with problems in stress management among patients with axSpA were disease activity assessed with BASDAI (OR 2.2, 95% CI: 1.6–3.3, *p* < 0.001) and depression (OR 18, 95% CI: 1.6–457.9, *p* = 0.03). Considering both populations together, patients with axSpA (OR 0.48, 95% CI: 0.24–0.95, *p* = 0.04) and those with severe disease (OR 6.6, 95% CI: 3.4–13.4, *p* < 0.001) were independently related to positive responses to at least one of the stress-handling items.

### 3.3. Frustration

Thirty-one of the 90 patients (34.4%) with PsA and 44/111 (39.6%) of the patients with axSpA reported a positive response to item #13. In the PsA group, there was a high correlation between item #13 and the sum score of the ASAS HI (Pearson’s r: 0.72). When the correlations between item #13 and the rest of the components of ASAS HI were compared, the highest correlation occurred with item #17 (r: 0.52). In the multivariate analysis, arthritis duration (OR 1.08, 95% CI: 1.01–1.18, *p* < 0.05) and disease activity (OR 1.27, 95% CI: 1.16–1.42, *p* < 0.001) were independently associated with a positive response to item #13. A severe disease according to the DAPSA score (OR 11.6, 95% CI: 3.6–43.2, *p* < 0.001) was also found to be independently related to a positive response to item #13. Patients with positive responses to item #13 had, on average, 1.22 (95% CI: 0.33–2.11, *p* = 0.008) points more in the PsAID questionnaire than patients who responded negatively to this item. In the axSpA group, there was a moderate correlation between item #13 and the sum score of the ASAS HI (r: 0.65). In this group, the strongest correlation between this item and the rest was found with #3 (“I have problems running”, r: 0.47), referred to the und ICF category of moving around. In the axSpA group, age (OR 0.95, 95% CI: 0.91–0.99, *p* < 0.05) and disease activity (OR 1.76, 95% CI: 1.41–2.28, *p* < 0.001) were found to be independently related to positive responses to item #13. In addition, disease severity according to BASDAI (OR 7.4, 95% CI: 2.9–19.6, *p* < 0.001) and ASDAS (OR 5.4, 95% CI: 2.2–13.8, *p* < 0.001) was associated with positive responses to this item. When both populations were analyzed together, only disease severity (OR 6.6, 95% CI: 3.4–13.1, *p* < 0.001) was independently associated with feelings of frustration. Figure 1 illustrates the distribution of the different ASAS HI items between both populations.

## 4. Discussion

In this study, a third of the patients with axSpA and a percentage close to 40% of the patients with PsA answered positively to at least one of the items of the ASAS HI referring to stress handling. Both stress-handling items of the ASAS HI had moderate correlations with the sum score of the ASAS HI. As a common point, disease activity was independently associated with positive responses to stress items in the two diseases. However, there were some differences in other aspects related to these responses. Thus, the relationships between biological therapy and the stress-related items showed opposite associations (negative in PsA and positive in axSpA). When we analyzed patients with at least one positive response (to either item #11 or 17), we found that patients with axSpA reduced this possibility by more than 50%, while patients with severe disease multiplied this option by more than 6.

We found that the activity and severity of both SpA subtypes were associated with a greater probability of expressing feelings of frustration. In the PsA group, arthritis duration was positively associated with item #13, while an inverse relationship was found between age and this response in the axSpA group. In the PsA group, item #13 showed the highest correlation between any of the items of the ASAS HI and its sum score. Furthermore, there was a good correlation between this response and an item referring to stress handling (#17). These findings highlight the importance of psychological factors as key drivers of HRQoL in PsA. The best correlation between items of the ASAS HI in the axSpA group occurred between #13 and #3 (which referred to the ICF category of moving around), which underscores that QoL relies more on physical than psychological factors in axSpA.

Spondyloarthritis has a negative impact on patients’ HRQoL, and it affects both physical and mental dimensions [1,9]. However, the correlations between disease variables and disease impact are only partially known in these conditions [1,9,13]. In the PsA group, a high disease activity score, chronic comorbidity, and radiographic damage negatively affected the physical components of HRQoL, while the severity of psoriatic lesions was significantly associated with a higher impact on the mental dimension [9,13,14]. In the axSpA group, the clinical variables were mainly correlated with physical components, whereas in the PsA group, both physical and mental components were affected by the disease [1,9,13,14]. Consistently with these findings, we found a strong link of disease activity and its severity with positive responses to stress-related items. On the other hand, although depression is usually more prevalent in PsA than in axSpA [1,4], this factor was only found to be associated with positive responses to items #11 and/or #17 in patients with axSpA.

Biological therapy has been associated with different positive HRQoL outcomes in patients with axSpA and PsA [15,16]. In our study, we found both positive (axSpA) and negative (PsA) associations in relation to the use of biological therapies and positive statements to stress-related components of the ASAS HI. Although the rates of use of this type of drug were different between the two entities, this was included in the multivariate analyses. There is, therefore, no clear explanation for these findings, especially if we consider that patients with axSpA had reduced chances of having any of the stress-related responses by more than 50% when both populations were analyzed together. We also found a clear link between inflammatory activity, as well as disease severity, and the altered emotional functions represented by item #13 of the ASAS HI. In that sense, recent evidence has suggested an inflammatory etiology of the psychological dysfunction in inflammatory arthritis [17,18]. Thus, pro-inflammatory mediators, such as interleukin (IL)-6, IL-17, and tumor necrosis factor (TNF)-α, have been associated with depression in SpA [17,18]. These relationships offer the possibility of addressing these comorbidities by targeting these pro-inflammatory cytokines.

Quality-of-life assessment tools do not always reflect patients’ views and perceptions regarding disease impact. Some QoL questionnaires, such as the Short Form-36 (SF-36) mental component summary and the Psoriatic Arthritis Quality of Life (PsAQoL), may be difficult for most physicians to implement in routine clinical practice due to their complexity [19]. In this sense, the ASAS HI offers the advantage of a simple, validated, and easily implemented tool in clinical routines to address these disease domains [20]. Therefore, the detection of psychological distress factors is relevant insofar as these factors can be associated with not only worse QoL, but also poorer survival in biological therapies [18,19].

Some weaknesses of this study should be highlighted. The first is that the ASAS HI is an instrument to be interpreted as a whole, not in parts. Moreover, both study populations were relatively small, and therefore, not all of the phenotypes of both processes were considered. Furthermore, the analysis of the stress-related items and that of emotional functions were not contrasted with more standard instruments. This means that the construct validity of these ASAS HI items is unknown. In addition, this analysis took the three items as a secondary and not as a primary outcome, so there were no adequate calculations of the sample size. The concept of severity used in this study (adding the activity metrics to the opinion of the evaluating physician) is not standard, and this can be criticized. On the other hand, the aspects related to psychosocial health analyzed in this study may be influenced by the concomitant presence of fibromyalgia or widespread chronic pain of other origins. For all of these reasons, our results need further validation and should be taken with caution. Finally, we should keep in mind that emotional functions are linked to a very wide range of psychosocial concepts (anxiety, fear, uncertainty, sadness, depression, lack of motivation, anger, etc.) beyond stress and frustration, so we still need to explore these SpA-related aspects in more detail.

The novelty of this study consisted in being able to analyze an aspect of the QoL of these patients that is little addressed in clinical practice (psychosocial health) by using a simple tool that is easy to implement in a clinical routine, namely, the ASAS HI (20).

In summary, at least a third of the study population reported problems related to stress handling and feelings of frustration. Both the activity and the severity of the disease were associated with these psychosocial issues in our context. Moreover, psychosocial factors seem to have a greater impact on HRQoL in PsA patients than in axSpA patients. Therefore, the detection and treatment of these factors should be part of a patient-centered approach for both diseases.

## Figures and Tables

**Figure 1 jcm-11-06094-f001:**
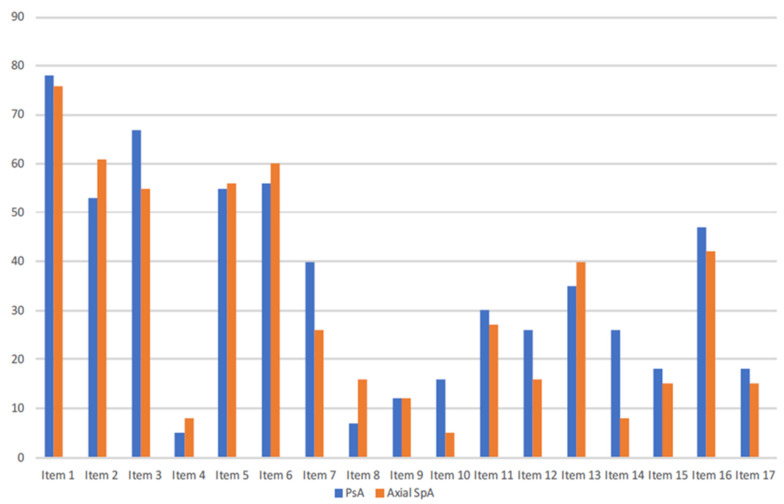
The distribution of the different components of the ASAS HI between both SpA subpopulations. Item 1: Pain sometimes disrupts my normal activities; Item 2: I find it hard to stand for long; Item 3: I have problems running; Item 4: I have problems using toilet facilities; Item 5: I am often exhausted; Item 6: I am less motivated to do anything that requires physical effort; Item 7: I have lost interest in sex; Item 8: I have difficulty operating the pedals in my car; Item 9: I am finding it hard to make contact with people; Item 10: I am not able to walk outdoors on flat ground; Item 11: I find it hard to concentrate; Item 12: I am restricted in traveling because of my mobility; Item 13: I often get frustrated; Item 14: I find it difficult to wash my hair; Item 15: I have experienced financial changes because of my rheumatic disease; Item 16: I sleep badly at night; Item 17: I cannot overcome my difficulties. Columns represent percentages of affirmative responders. ASAS HI: Assessment of Spondyloarthritis International Society–Health Index; PsA: psoriatic arthritis; AxSpA: axial spondyloarthritis.

**Table 1 jcm-11-06094-t001:** Main disease features of both study populations.

Axial Spondyloarthritis	N: 111	Psoriatic Arthritis	N: 90
Age, yrs (mean ± SD)	43.3 ± 10.6	Age (yrs), (mean ± SD)	55.3 ± 14.3
Disease duration, yrs (mean ± SD)	7.6 ± 6.8	Arthritis duration (yrs), median (IQR) Psoriasis duration (yrs), median (IQR)	7 (3–14) 16 (10–29)
Men, n (%)	74 (66.7)	Men, n (%)	52 (57.7)
AS, n (%) Peripheral involvement, n (%)	74 (66.7) 18 (16.2)	Plaque psoriasis, n (%) Nail disease, n (%) ≥3 Affected body areas, n (%)	75 (83.3) 33 (36.7) 61 (67.8)
Family history, n (%)	16 (14.4)	Family history of psoriasis, n (%) Family history of PsA, n (%)	37(41.1) 6 (6.7)
Primary education, n (%) Secondary education, n (%) University degree, n (%)	43 (38.7) 34 (30.6) 34 (30.6)	Primary education, n (%) Secondary education, n (%) University degree, n (%)	25 (27.8) 45 (50) 20 (22.2)
Tobacco, n (%) Obesity, n (%) Hypertension, n (%) Diabetes, n (%) Dyslipidemia, n (%) Cardiovascular adverse events, n (%)	44 (39.6) 18 (16.2) 14 (12.6) 6 (5.4) 26 (23.4) 1 (0.9)	Diabetes, n (%) Hypertension, n (%) Dyslipidemia, n (%) Obesity, n (%) Smokers, n (%) Adverse cardiovascular events, n (%)	13 (14.4) 37 (41.1) 30 (33.3) 21 (23.3) 14 (15.6) 6 (6.7)
Enthesitis, n (%) Anterior uveitis, n (%) Inflammatory bowel disease, n (%)	8 (7.2) 14 (12.6) 6 (5.4)	Axial pattern, n (%) Oligoarthritis, n (%) Polyarthritis, n (%)	14 (15.6) 42 (46.7) 34 (37.8)
HLA-B27, n (%)	88 (79.3)		
NSAID use, n (%) Biologic therapy, n (%)	89 (80.2) 67 (60.4)	Conventional DMARDs, n (%) Biological therapy, n (%)	81 (90) 37 (41.1)
BASDAI, mean (SD) ASDAS-CRP, mean (SD) ASAS-HI, mean (SD)	3.4 (2.3) 2.1 (0.8) 5.4 (3.8)	DAPSA, mean (SD) PsAID, mean (SD) ASAS-HI, mean (SD)	9.7 (7.8) 2.8 (2.3) 5.8 (4.3)

yrs: years, SD: Standard Deviation, n: numbers, AS: Ankylosing Spondylitis, HLA: Human Leukocyte Antigen. NSAID: Non-Steroidal Anti-Inflammatory Drugs, BASDAI: Bath Ankylosing Spondylitis Disease Activity, ASDAS-CRP: Ankylosing Spondylitis Disease Activity Score–C-Reactive Protein, ASAS-HI: Assessment of SpondyloArthritis international Society–Health Index, IQR: InterQuartile Range, PsA: Psoriatic Arthritis, DMARDs: Disease Modifying AntiRheumatic Drugs, DAPSA: Disease Activity index for PSoriatic Arthritis, PsAID: Psoriatic Arthritis Impact of Disease.

## Data Availability

All of the data underlying the results are available as part of the article and its corresponding bibliographic references [11,12], so no additional source data are required.

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
