# Peer review of "Non-Physical Disease Facets in Spondyloarthritis: An ASAS Health Index-Based Analysis between Psoriatic Arthritis and Axial Spondyloarthritis"

_jcm, 2022, doi:10.3390/jcm11206094_

Round 1
Reviewer 1 Report
Table must be improved. Is it possible that age median in psoriatic arthrtitis grop was 55 and psoriasis duration 53 years?
Author Response
We thank the reviewer for her/his comments. Indeed, the quality of the table can be improved. A new table is presented in the new version of the manuscript.
Reviewer 2 Report
Psychosocial factors are well recognised in musculoskeletal disease and are an important driver of symptoms. This paper has a useful aim of trying to tease out the impact of these symptoms on quality of life.
I have a number of comments
1 In the introduction you state that around 2/3 patients in other studies report moderate/major impact on emotional wellbeing, yet only around a third in this study have reported this. Is there an explanation for this discrepancy
2. There is an assumption that the 3 questions chosen in this analysis are the only questions in the ASAS HI which assess emotional wellbeing. However, many of the questions could be interpreted as having emotional aspects (5/6/7/9/13/16/17). Why were these 3 qusestions chosen? There is often a complex interplay between emotional physical health and a gross yes/no answer as used in the HI may not be useful enough to gauge responses.
3. The formatting of table 1 makes this extremely difficult to interpret - I have to say I gave up.
4. you mention an association with item 11 for both disease activity and severity, yet these are all defined by disease activity measures. Have you distinguished these aspects in different ways. If so, then this is unclear.
5. I am not sure that the correlation matrices are helpful as supplementary information. Are there legends missing? Did you correct for multiple testing?
6. The yes/no format of the items in the HI make these data difficult to interpret as will capture mild/moderate and severe symptoms in a single value. Were other items collected that made greater sense of the magnitude of impairment?
Minor points
First line of introduction - Spondyloarthropathies (pleural) rather than spondyloarthropathy (singular). This error recurs later in the text.
Paragraph 2 (introduction) Many clinicians do ask about psychosocial health; I would suggest a change to less commonly rather than not usually.
A discussion regarding the co-existence of fibromyalgia/chronic pain syndromes in both axspa and psa may be helpful.
Author Response
1 In the introduction you state that around 2/3 patients in other studies report moderate/major impact on emotional wellbeing, yet only around a third in this study have reported this. Is there an explanation for this discrepancy
R/Thank you very much for this point. What we wanted to present in the introduction is that psychosocial health is frequently affected in this population. For this, we have put only a few examples of this point taken from the recent literature. It should be borne in mind that these works are basically surveys sent to a significant number of patients, while our work was only a point estimate over time using the ASAS HI. I think the reason for the discrepancy lies in the different methodology of those studies compared to ours.
- There is an assumption that the 3 questions chosen in this analysis are the only questions in the ASAS HI which assess emotional wellbeing. However, many of the questions could be interpreted as having emotional aspects (5/6/7/9/13/16/17). Why were these 3 qusestions chosen? There is often a complex interplay between emotional physical health and a gross yes/no answer as used in the HI may not be useful enough to gauge responses.
R/ We completely agree with this point. As we mentioned among the weaknesses of the study, the ASAS HI was designed for its interpretation as a whole, and not so much for the analysis of its individual components. Essentially, our objective focused on very partial aspects of psychosocial health, and for this reason we only chose two of the ICF categories referring to this aspect (frustration and stress management), anticipating that this is a very partial view of the whole problem, as rightly pointed out by the reviewer. Therefore, this is a merely exploratory study that requires further support through larger studies and with other psychosocial health measurement instruments.
- The formatting of table 1 makes this extremely difficult to interpret - I have to say I gave up.
R/ We provide now a new table
- you mention an association with item 11 for both disease activity and severity, yet these are all defined by disease activity measures. Have you distinguished these aspects in different ways. If so, then this is unclear.
R/ Again, thank you very much for the comment. Indeed, disease activity was evaluated by conventional metrics (BASDAI, ASDAS-PCR, DAPSA), and for severity (which is a different and more complex construct) the opinion of the evaluating physician was added to the activity variables, through a VAS 0-10 (arbitrarily choosing a cut-off point of 4, so that above this value, the doctor considered that the patient had a severe disease). We understand criticism, since it is a somewhat atypical way of evaluating the severity of an illness. In summary, high activity was assessed by the cutoff points of conventional instruments, and for severity, high activity was added to the opinion of severity expressed by the evaluating physician. We added a comment about it in the discussion.
- I am not sure that the correlation matrices are helpful as supplementary information. Are there legends missing? Did you correct for multiple testing?
R/ We agree. Supplementary material has been removed. The idea of incorporating the correlation matrices was to offer the interested reader more information about it, but we agree with the reviewer that this material can be ignored.
- The yes/no format of the items in the HI make these data difficult to interpret as will capture mild/moderate and severe symptoms in a single value. Were other items collected that made greater sense of the magnitude of impairment?
R/ Again very good point. As we have commented in the section on the weaknesses of the study, the ASAS HI is a continuous measure designed for its global interpretation, so that the higher the score, the greater the impact on quality of life. For this reason, the sub-analysis of some of its components can give a partial idea of what the questionnaire really intends to measure. Also, as we mentioned in the section on the weaknesses of the study, psychosocial health admits many more constructs than just the feeling of frustration or stress management problems, so we insist on the cautious interpretation of our findings.
Minor points
First line of introduction - Spondyloarthropathies (pleural) rather than spondyloarthropathy (singular). This error recurs later in the text.
R/ Corrected.
Paragraph 2 (introduction) Many clinicians do ask about psychosocial health; I would suggest a change to less commonly rather than not usually.
R/ Corrected following your suggestion.
A discussion regarding the co-existence of fibromyalgia/chronic pain syndromes in both axspa and psa may be helpful.
R/ Thank you again for this suggestion. Included in the discussion section.
Round 2
Reviewer 2 Report
Previous comments have now been addressed; thank you. A final formatting/grammar check may be helpful.